# Pilocarpine inhibits *Candida albicans SC5314* biofilm maturation by altering lipid, sphingolipid, and protein content

Emerenziana Ottaviano,[1] Michele Dei Cas,[1] Silvia Ancona,[1] Francesca Triva,[1] Sara Casati,[2] Francesca Sisto,[2] Elisa Borghi[1]

**ABSTRACT** *Candida albicans* filamentation and biofilm formation are key virulence factors tied to tissue invasion and antifungal tolerance. Pilocarpine hydrochloride (PHCl), a muscarinic receptor agonist, inhibits biofilm maturation, although its mechanism remains unclear. We explored PHCl effects by analyzing sphingolipid and lipid composition and proteomics in treated *C. albicans* SC5314 biofilms. PHCl significantly decreased polar lipid and ergosterol levels in biofilms while inducing phytoceramide and glucosylceramide accumulation. PHCl also induced reactive oxygen species and early apoptosis. Proteomic analysis revealed that PHCl treatment downregulated proteins associated with metabolism, cell wall remodeling, and DNA repair in biofilms to levels comparable to those observed in planktonic cells. Consistent with ergosterol reduction, Erg2 was found to be reduced. Overall, PHCl disrupts key pathways essential for biofilm integrity, decreasing its stability and promoting surface detachment, underscoring its potential as a versatile antifungal compound.

**IMPORTANCE** *Candida albicans* filamentation and biofilm formation represent crucial virulence factors promoting fungus persistence and drug tolerance. The common eukaryotic nature of mammalian cells poses significant limitations to the development of new active nontoxic compounds. Understanding the mechanism underlying PHCl inhibitory activity on yeast–hypha transition, biofilm adhesion, and maturation can pave the way to efficient drug repurposing in a field where pharmaceutical investment is lacking.

**KEYWORDS** *Candida albicans*, biofilms, drug repurposing

*C*andida albicans is recognized as one of the most prevalent human fungal pathogens, responsible for a wide spectrum of clinical conditions, ranging from superficial infections of the oral and vaginal mucosa to life-threatening systemic diseases in immunocompromised patients (1).

The ability of *C. albicans* to form highly structured biofilms, consisting of various cell types surrounded by an extracellular matrix, is a crucial virulence factor leading to tissue invasion and antifungal tolerance (2). *Candida* biofilms have been observed on most used medical devices, such as shunts, endotracheal tubes, and catheters (2). Implanted medical devices provide the necessary surfaces for yeasts to form a complex network of biofilm. Biofilms serve as reservoirs of cells to continuously seed an infection and are currently responsible for a significant percentage of candidiasis (3). Biofilm-forming isolates are able to cause experimental invasive infections with a higher mortality rate (4, 5). Indeed, microbial biofilms are difficult for immune cells to recognize and engulf and display tolerance to antimicrobial therapies (6, 7). Despite azoles and echinocandins being the first-choice drugs for candidiasis treatment, the emergence of drug resistance and the inherent pharmacological tolerance of the sessile form of *Candida* underscore the urgency to develop new antifungal strategies (8, 9).

**Peer Reviewer** Walid Salem Abu Rayyan, Al-Zaytoonah University of Jordan, Amman, Jordan

Address correspondence to Elisa Borghi, elisa.borghi@unimi.it.

The authors declare no conflict of interest.

In recent years, drug repurposing has gained traction for treating drug-resistant infections. The use of pilocarpine hydrochloride (PHCl) against *C. albicans* biofilm-related infection has been recently investigated (10), stemming from data on acetylcholine that inhibits yeast filamentation and triggers an efficient and nondetrimental immune response (11). PHCl, a licensed cholinergic drug, strongly impairs *C. albicans* biofilm biomass and metabolic activity in a dose-dependent manner (10). Despite a muscarinic receptor being postulated to be involved in PHCl response, no clear mechanisms have been depicted to date.

As PHCl decreases *C. albicans* cell surface hydrophobicity that is related to cell wall composition, an effect on cell wall composition and stability could be involved in biofilm impairment.

In the present study, we investigated the modulatory effects of PHCl through sphingolipidomics, lipidomics, and proteomics approaches.

## MATERIALS AND METHODS

### Chemicals and reagents

The chemicals methanol, chloroform, formic acid, ammonium acetate, ammonium formate, dibutylhydroxytoluene (BHT), urea, ammonium bicarbonate, dithiothreitol (DTT), and iodoacetamide (IAA) were all at analytical grade and purchased from Sigma-Aldrich (St. Louis, MO, USA). All aqueous solutions were prepared using purified water at a Milli-Q grade (Burlington, MA, USA). Lipid standards were purchased from Avanti Polar (supplied by Sigma-Aldrich). $H_2O_2$ and PHCl hydrochloride 98% powder (Sigma-Aldrich) were dissolved in distilled water. Zymolyase, sorbitol (1 M), potassium phosphate (50 mM), and EDTA (5 mM) were purchased from Sigma-Aldrich.

### *Candida albicans* SC5314 culture conditions

The *C. albicans* SC5314 reference strain was grown overnight in YPD (yeast extract peptone dextrose) broth (LLG Labware) at 30°C with shaking. Stationary-phase cultures were washed twice in cold sterile phosphate-buffered saline (PBS; Sigma-Aldrich) and resuspended at $5 \times 10^6$ cells/mL in Roswell Park Memorial Institute (RPMI) 1640 medium (Sigma-Aldrich).

*Candida* cells were seeded in 90 mm Petri dishes to allow biofilm formation and in 15 mL tubes, under continuous shaking at 120 rpm (SHEL LAB shaking incubator) for planktonic growth, with or without 25 mM PHCl, and incubated for 24 hours at 37°C. Yeast cells were then harvested by centrifugation at 10,000 *g* for 10 minutes (Z 300K Universal Centrifuge, Hermle LaborTechnik GmbH), and the obtained pellets were lysed in PBS by bead-beating mechanical disruption at 4°C (TissueLyser LT, Qiagen). The supernatants, collected by centrifugation at 10,000 *g* for 10 minutes, were stored at −20°C until further use. Three different samples have been prepared for each condition, and the experiment has been conducted three times. Protein concentration was measured by bicinchoninic acid assay (BCA) (Pierce BCA Protein Assay Kit, Thermo Scientific) and used to normalize sphingolipidomic, lipidomic, and proteomic results.

### Untargeted lipidomics

Samples from *C. albicans* (100–200 µg protein) were resuspended in water (100 µL) and extracted with methanol/chloroform mixture (850 µL, 2:1, v/v) in an oscillator thermomixer (60 min, 1000 rpm, 4°C; ThermoCell Mixing Block, MB-102, Bioer Technology). After centrifugation (15 min at 17,000 *g*) (Z216MK, Hermle LaborTechnik GmbH), the organic phase was evaporated under a stream of nitrogen. The residues were dissolved in 100 µL of isopropanol/acetonitrile (2:1, v/v) + 0.1 mM BHT and transferred to a glass vial.

The instrument consisted of a Shimadzu UPLC coupled with a Triple TOF 6600 Sciex (Concord, Canada) equipped with a Turbo Spray Ion Drive. All samples were analyzed in duplicate in both positive and negative modes with electrospray ionization. The instrument settings were as follows: CUR = 35, GS1 = 55, GS2 = 65, capillary voltage

±5.5 kV, and source temperature 350°C. Spectra were contemporaneously acquired using a full-mass scan from $m/z$ 200 to 1,500 (100 ms accumulation time) and data-dependent acquisition from $m/z$ 50 to 1,500 (40 ms accumulation time, top-20 spectra per 0.8 s cycle). The declustering potential was fixed to 50 eV, and the collision energy (CE) was 35 ± 15 eV. The chromatographic separation was achieved on a reverse-phase Acquity CSH C18 column (1.7 µm, 2.1 × 100 mm; Waters, MA, USA) equipped with a precolumn, using mobile phase (A) water/acetonitrile (60:40) and mobile phase (B) 2-propanol/aceto-nitrile (90:10), both containing 10 mM ammonium acetate and 0.1% formic acid (8). The flow rate was 0.4 mL/min, and the column temperature was 55°C. The elution gradient was programmed from 40% to 90% B in 19 minutes. Five microliters of clear organic supernatant were directly injected in the LC-MS/MS. The spectra deconvolution, peak alignment, and sample normalization were attained using MS-DIAL (ver. 4.7). MS and MS/MS tolerance for peak profile was set to 0.01 and 0.05 Da, respectively. Identification was achieved by matching molecular ($m/z \pm 0.02$) and MS/MS experimental spectra ($m/z \pm 0.05$) with the LipidBlast library for lipidomics. Features meeting the following criteria were extracted and used for further analysis: (i) the CV% of the feature in the QC sample, repeatedly injected all along with the batch, should be <30%, and (ii) the value of the feature peak was more than 10-fold the value of the same feature in the blank. Intensities of the remaining metabolites were normalized by the Lowess algorithm. Fungal sterols were manually validated by mass spectral interpretation as follows: ergosterol (m/z 379.33603, rt 11.26, MS/MS 253.19, 125.13, 159.11, 295.24, 309.25), 5-dehydroergosterol (m/z 381.35001, rt 12.8, MS/MS 159.11, 213.16, 255.21), and their precursor lanosterol (m/z 409.38422, rt 14.51, MS/MS 327.30, 191.18, 121.10, 147.11, 159.11).

## Sphingolipidomics

Samples were extracted and analyzed for the evaluation of their sphingolipid profile that includes sphingolipids with different sphingoid bases such as sphingosine (d18:1), sphinganine (d18:0), 4-OH sphinganine (phytosphingosine, t18:0) and sphinga-4,8-dien-ine (d18:2), 4-OH eicosa-sphinganine (t20:0), and 9-methyl-sphinga-4,8-dienine (d18:2, 9Me or d19:2). Moreover, sphingolipids with alpha-hydroxy fatty acids (2OH) were also investigated.

Samples from *C. albicans SC5314* (50 µg protein) were resuspended in water (100 µL), added with IS (20 µM of Cer C12:0, HexCer C12:0, SM C12:0 in methanol), and extrac-ted with methanol/chloroform mixture (850 µL, 2:1, v/v) in an oscillator thermo-mixer (30 min, 1000 rpm, rt). They were subjected to alkaline methanolysis (75 µL KOH 1M in methanol) for 2 hours at 38°C and, at the end, neutralized by the addition of pure acetic acid (4 µL). After centrifugation (15 min at 15,000 g), the organic phase was evaporated under a stream of nitrogen. The residues were dissolved in 100 µL of methanol + 0.1 mM BHT and transferred to a glass vial.

They were analyzed by LC-MS/MS using a LC Dionex 3000 UltiMate LC system (ThermoFisher Scientific, Waltham, MA, USA) coupled to an AB Sciex 3200 QTRAP tandem mass spectrometer (AB Sciex, Concord, Canada) equipped with an electrospray ionization TurbolonSpray source operating in positive mode (ESI+). The instrument parameters were as follows: CUR 25, GS1 45, GS2 50, capillary voltage 5.5 kV, and source temperature 300°C. Spectra were acquired by multiple reaction monitoring, scanning the following mass fragments (MS/MS) for each analyte: $m/z$ 264.27 for ceramide and glycosphingolipids; $m/z$ 266.27 for dihydroceramide; $m/z$ 184.07 for sphingomyelins; $m/z$ 300.3 and 282.4 for glyco- and phytoceramide; $m/z$ 262.30 for glyco- and 4,8-diene-ceramide; and $m/z$ 276.30 for 9-methyl-4,8-diene-ceramide. Negative ionization (ESI−) was used to check the mass fragments (MS/MS): $m/z$ 241.0 and 259.1 for IPC (CE −65 ev) and $m/z$ 241.0, 403.1, and 421.07 for MIPC (CE −80 eV). Precursor ions for each sphingoli-pid were found on LIPID MAPS (https://www.lipidmaps.org/databases/lmsd/browse) or calculated if missing.

Chromatographic separation was achieved on a reverse-phase Acquity BEH C8 column (1.7 µm, 2.1 × 100 mm; Waters, MA, USA) equipped with pre-column, using the

following mobile phases: (A) water + 0.2% formic acid + 2 mM ammonium formate and (B) methanol + 0.2% formic acid + 1 mM ammonium formate. The flow rate was 0.3 mL/min, and the column temperature was set to 30℃. The elution gradient (%B) was set as follows: 0–3 min (80–90%), 3.0–6.0 min (90%), 6.0–19.0 min (90–99%), and 19.0–20.0 min (99–80%), held until 24 minutes. Five microliters of clear supernatant were directly injected into LC-MS/MS. Inositol phosphoceramide (IPC) and mannosyl-IPC (MIPC) were also analyzed by a high-sensitivity LC–MS/MS consisting of a QTrap 5500 triple quadrupole linear ion trap mass spectrometer (Sciex, Darmstadt, Germany) equipped with an electrospray ionization (ESI) source operating in negative ionization mode and coupled with an Agilent 1200 Infinity pump Ultra High-Pressure Liquid Chromatography (UHPLC) system (Agilent Technologies, Palo Alto, CA, USA). Chromatographic separation was carried out as previously described, and the mass fragments (MS/MS) monitored were $m/z$ 241.0 and 259.1 for IPC (CE 65 eV) and $m/z$ 241.0, 403.1, and 421.07 for MIPC (CE 80 eV).

## Label-free proteomics

Samples were resuspended in PBS, mechanically lysed by shaking in a TissueLyser (2 cycles at 50 Hz per 5 min), ice sonicated for 30 minutes, and proteins denatured in 8M urea, tris HCl (pH 8.5) + 0.1% protease inhibitor (cOmplete Protease Inhibitor Cocktail, Roche). Samples were then diluted in 50 mM ammonium bicarbonate (pH 8.2), reaching a nominal concentration of 1 µg/µL. 10 µg of each sample were processed for digestion after reduction (DTT 100 mM, 55° per 30 min) and alkylation (IAA 300 mM, 20 min in the dark) of cysteine residues. Samples were digested overnight at 37℃ by adding trypsin at a 1:40 ratio (0.25 µg per sample). They were brought to pH <2 by the addition of 1 µL of pure TFA. Proteins were desalted and purified by using ZipTip C18 (100 µg) according to manufacturer's instructions. The eluates were evaporated under nitrogen and resuspended in 10 µL of 0.1% formic acid (FA).

All samples were analyzed at UNITECH OMICs (Università degli Studi di Milano, Italy) using Dionex Ultimate 3000 nano-LC system (Sunnyvale CA, USA) connected to Orbitrap Exploris 240 Mass Spectrometer (Thermo Scientific, Bremen, Germany) equipped with nano-electrospray ion source. Peptide mixtures were pre-concentrated onto an Acclaim PepMap 100–100 µm × 2 cm C18 (Thermo Scientific) and separated on an EASY-Spray column ES900 (15 cm × 75 µm ID) packed with Thermo Scientific Acclaim PepMap RSLC C18, 3 µm, 100 Å, using mobile phase A (0.1% formic acid in water) and mobile phase B (0.1% formic acid in acetonitrile 20/80, v/v) at a flow rate of 0.300 µL/min. The elution program spanned from 4% to 95% B in 150 min. The temperature was set to 35℃, and the sample was injected in duplicates. The sample injection volume is 3 µL.

Data elaboration was completed with Proteome Discoverer 2.5, equipped with *C. albicans* database (sp_incl_isoforms TaxID = 5476_and_subtaxonomies; (v2022-06-14)).

## Flow cytofluorometric analysis

Flow cytometric analysis (FACS) was performed on *C. albicans* planktonic cells. Yeast cells were grown in YPD broth overnight under shaking conditions, and the next day subcultured (1:20) in 3 mL fresh YPD broth, grown for 3 hours, and then treated overnight at 37℃ with 250 µM $H_2O_2$ in water (positive experimental control) or 25 mM PHCl (Sigma Aldrich, Saint Louis, MO, USA), and treated with PBS (negative control). Prior to FACS analysis and specific staining, yeast cells were incubated with zymolyase in Sorbitol 1 M buffer, potassium phosphate (50 mM), EDTA (5 mM) for 30 min at 37℃ and washed twice in PBS 1× to obtain spheroplasts. For apoptosis analysis, cells were then incubated for 15 min at room temperature with APC Annexin V (Biotium, Inc., Fremont, CA, USA) and PI (ThermoFisher Scientific, Waltham, MA, USA) in an appropriate volume of Annexin V Binding Buffer (ThermoFisher Scientific) and washed once with PBS 1X. For ROS evaluation, cells were incubated for 30 min at 37℃ with dihydroethidium (DHE) and washed once with PBS 1×.

Samples were analyzed using FACSVerse (Beckman Coulter, Brea, CA, USA). Data acquisition and analysis were done using CXP 2.2 software (Beckman Coulter).

## Statistics

Statistical analyses were performed using GraphPad Prism 10 (GraphPad software, La Jolla, CA, USA). Data are expressed as mean ± SE, calculated from experimental replicates. For lipidomic and sphingolipid analyses, data significance was evaluated using two-way ANOVA followed by Bonferroni correction ($P < 0.05$) or Mann-Whitney test, as detailed in the figure legends. Statistical analysis for proteomics was elaborated using Proteome Discoverer 2.5; the $P$ value for pairwise protein ratio was investigated by background-based $t$-test.

In all analyses, $P < 0.05$ was considered statistically significant.

## RESULTS

### PHCl decreases polar lipid content in sessile cells

Lipid content differs between sessile and planktonic *Candida* cells, and these changes can affect fungal pathogenesis and antifungal resistance (12–14). Untargeted lipidomic analysis allowed the identification of monoacylglycerols (MGs), diacylglycerols (DGs), and triacylglycerols (TGs) as well as different classes of phosphoglycerides (PGLs), namely, phosphatidylcholine (PC, 1,2-diacyl-sn-glycero-3-phosphocholine), lysophosphatidylcholine (LPC, 1-acyl-sn-glycero-3-phosphocholine), phosphatidylethanolamine (PE, 1,2-diacyl-sn-glycero-3-phosphoethanolamine), lysophosphatidylethanolamine (LPE, 1-acyl-sn-glycero-3-phosphoethanolamine), and phosphatidylinositol (PI, 1,2-diacyl-sn-glycero-3-phospho-(1'-myo-inositol).

When comparing planktonic and sessile cells, we detected in the latter a generalized increase in lipid content except for lanosterol and monoacylglycerols (Fig. 1). Notably, we observed an enrichment of polar lipids in sessile cells, including PC (3.08-fold change compared with planktonic cells), PE (2.74-fold change), and PI (4.49-fold change). The PC:PE ratio was also increased compared to planktonic cells. DGs and TGs were also increased (2.9-fold).

In contrast, 24-hour treatment with PHCl dramatically reduced the lipid content in sessile cells, not only compared with untreated biofilms but to their planktonic counterparts. We observed a consistent decrease in PC (15-fold change compared to untreated sessile cells), in PE (10-fold change), and PI (20-fold). LPC and LPE, almost unaltered between planktonic and sessile status, were also decreased by PHCl (4- and 2-fold, respectively). On the other hand, PHCl dramatically enhanced lanosterol content in sessile cells, with a concomitant depletion in the ergosterol and the late sterol intermediate dehydroergosterol.

### Sphingolipid content varies between planktonic and sessile and PHCl-treated cells

Sphingolipid (SL) biosynthesis and metabolism (Fig. 2) are critical for a variety of cellular homeostatic processes, with different SL classes involved in yeast-specific pathways (15).

We analyzed and compared SL moieties in planktonic and sessile (biofilm-organized) *C. albicans* cells, treated or not with PHCl 25 mM (Fig. 3).

Ceramide content was strongly reduced in both sessile cells and PHCl-treated planktonic cells compared to untreated yeast cells ($P = 0.0017$ and $P = 0.0008$, respectively), whereas no significant differences were found for dihydroceramide (DHCer). Glucosylceramides (GlcCer), mostly derived from ceramide, were significantly increased by PHCl treatment in planktonic cells and to a lesser extent in biofilm, whereas the hydroxylated forms, α-OH GlcCer, were statistically enriched by PHCl in both conditions ($P = 0.0059$ for planktonic and $P = 0.0292$ for sessile cells).

PHCl treatment also increased phytoceramide (PhytoCer) production in both planktonic and biofilm-organized cells, with a greater effect on the latter ($P = 0.0105$ and

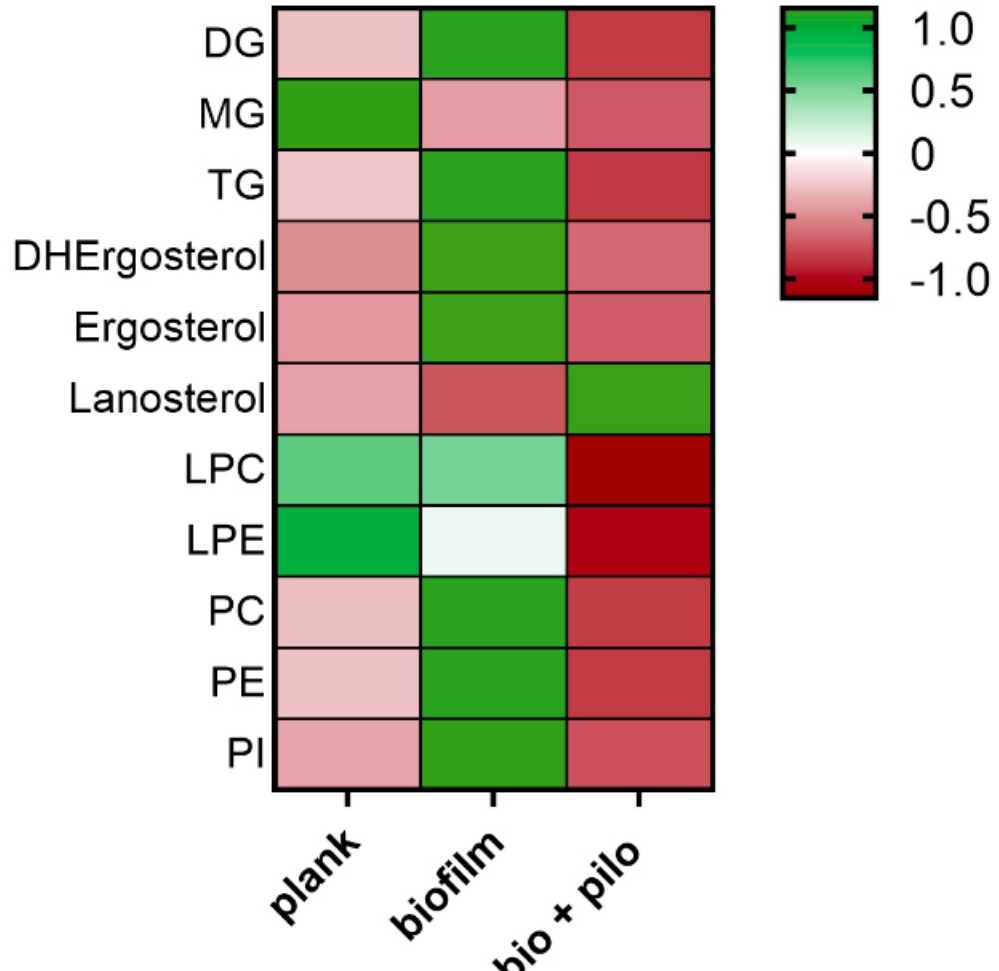

**FIG 1** Lipid classes in planktonic and sessile *C. albicans*, treated or not for 24 hours with PHCL 25 mM. Values are represented as *z*-score after being normalized for the protein content of each sample. DG, diacylglycerol; MG, monoacylglycerol; TG, triacylglycerol; DHErgosterol, dehydroergosterol; LPC, lysophosphatidylcholine; LPE, lysophosphatidylethanolamine; PC, phosphatidylcholine; PE, phosphatidylethanolamine; PI, phosphatidylinositol.

$P = 0.0002$, respectively). The hydroxylated form, α-OH PhytoCer, content was increased by PHCl only in biofilms ($P < 0.0001$). Inositol phosphoryl ceramides (IPCs), mostly derived from DHCer and PhytoCer and precursors of mannosylated IPCs, increased during biofilm formation ($P = 0.0475$) but were not affected by PHCl treatment. We did not find detectable amounts of mannosylated complex sphingolipids (MIPC and MIP$_2$C).

## Biofilm formation and PHCl treatment induce proteomic remodeling in *C. albicans*

To explore possible PHCl mechanisms involved in biofilm inhibition, we evaluated proteomic changes in *C. albicans* cells induced by the transition from planktonic to biofilm status (Fig. 4, upper donut charts) and by PHCl treatment in sessile cells (Fig. 4, lower donut charts).

We have identified over 500 proteins, 16 of which were significantly upregulated and 21 were downregulated in 24-hour-old biofilms compared to the planktonic cells, indicating a deep reprogramming of the proteome during biofilm formation. Four proteins were solely expressed in sessile cells, i.e., Ett1, Bgl2, Rfx2, and Def1 ($P < 0.0001$). Among the upregulated ones, we identified several proteins associated with hyphal growth, such as Hyr1 (hyphal-regulated cell wall protein 1, $P < 0.0001$), Ihd1 (induced

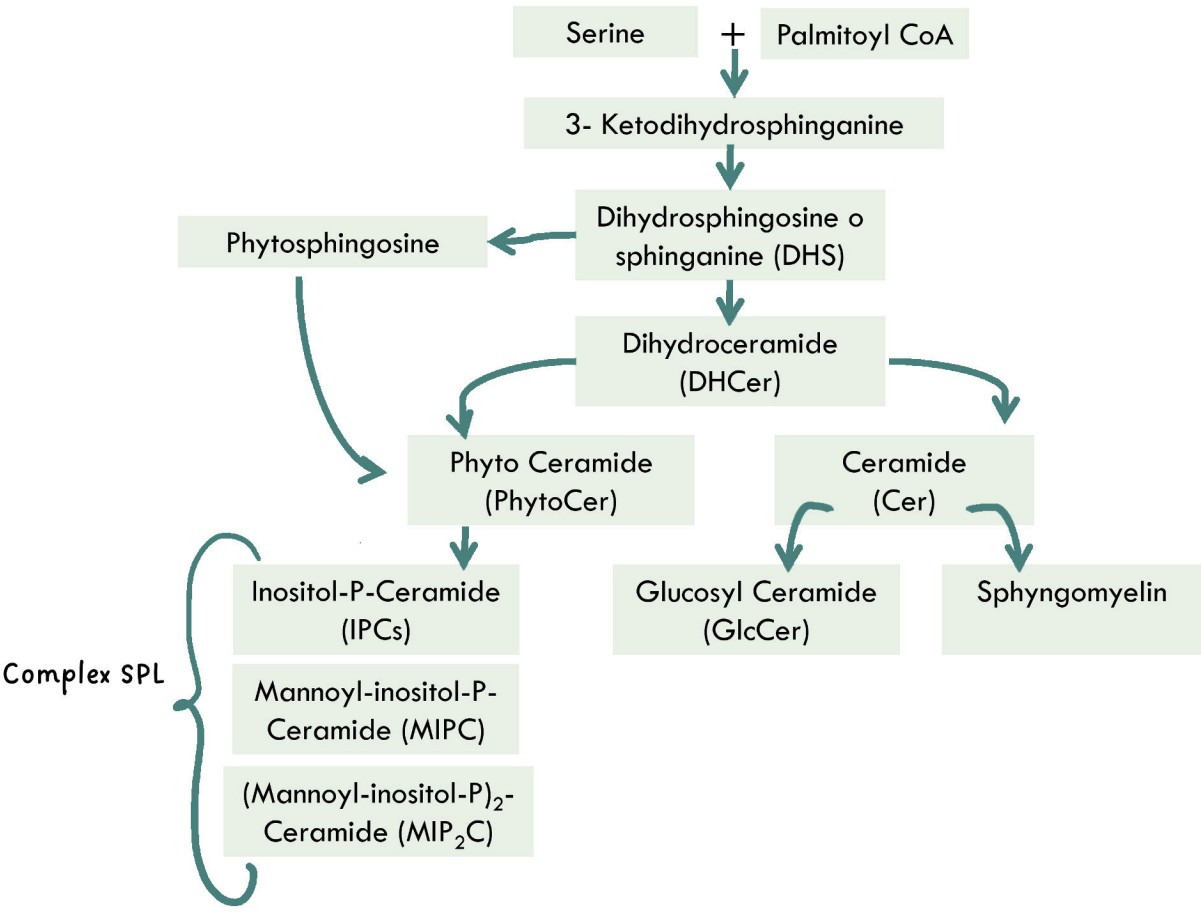

**FIG 2** Scheme of sphingolipid biosynthesis in fungi.

during hyphae development protein 1, $P = 0.0011$), adhesion, i.e., Als3 (agglutinin-like sequence 3, $P < 0.0001$), and chitin remodeling (Cht2, $P < 0.0001$).

Proteomic data from PHCl-treated sessile cells showed 12 statistically significant upregulated and 30 downregulated hits. PHCl induced the exclusive expression ($P < 0.0001$) of proteins that were also increased in planktonic cells such as Eng1 (endo-1,3, glucanase 1), Efg1 (rRNA-processing protein, Accession Q5AK42), Rbe1 (repressed by Efg1 protein 1), and Rsa3 (ribosome protein assembly 3). PHCl treatment also upregulated some histone proteins (H2A and H2B) and metabolic enzymes (isocitrate lyase, sorbose reductase, dihydrofolate reductase, and inositol-3-phosphate synthase). Several proteins belonging to the translational machinery were also found to increase by PHCl (Rix1, L39, P2-b, and CaYST1).

## PHCl induces reactive oxygen species production leading to *C. albicans* apoptosis

Drug resistance and drug repurposing strategies are shared topics in cancer and infection research. Since PHCl has been shown to induce reactive oxygen species (ROS) production, leading to mitochondrial dysfunction in human hepatoma-derived HepG2 cells (16), we investigated whether PHCl inhibition of biofilm formation and growth (10) could involve an increase in ROS levels and cell death. To this end, we evaluated the ROS status in *C. albicans* after 24 hours of PHCl 25 mM exposure by measuring cytosolic superoxide and ONOO or •OH production by dihydroethidium (DHE). We found that PHCl induced a significant increase in ROS production ($P = 0.0205$) compared to untreated cells (Fig. 5A).

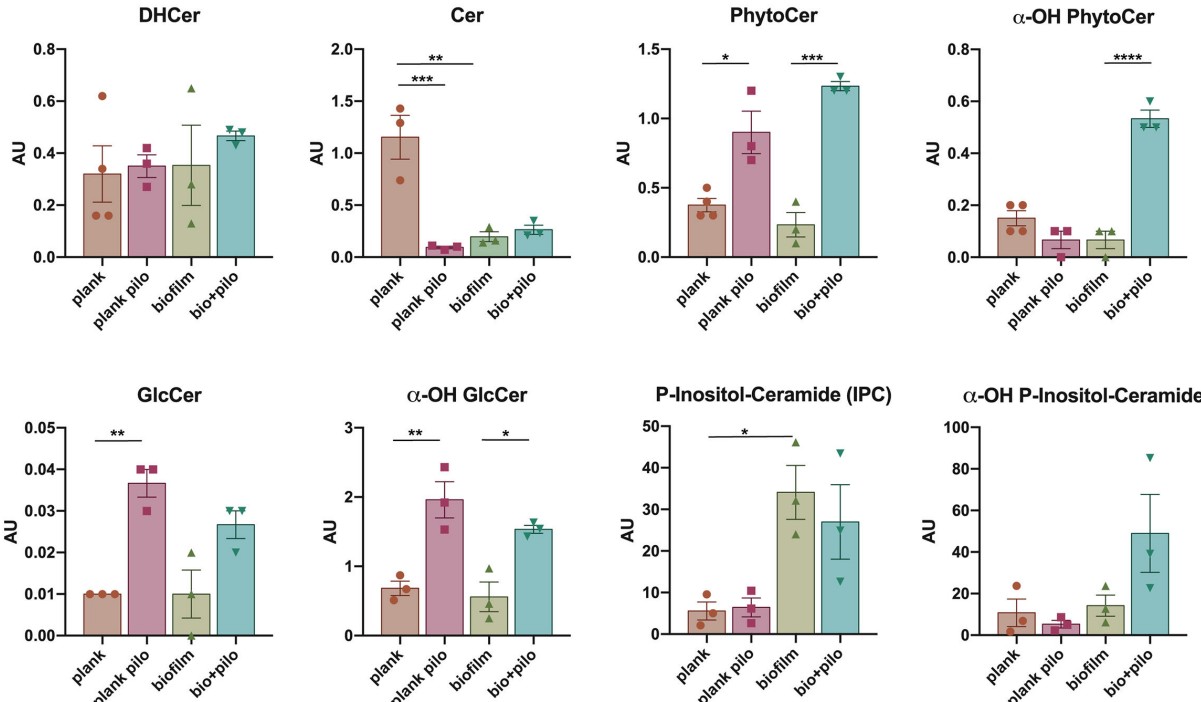

**FIG 3** Sphingolipid classes in planktonic and sessile *C. albicans*, treated or not for 24 hours with PHCL 25 mM. Statistical analysis by ANOVA; *P* < 0.05; \*\**P* < .01; \*\*\**P* < .001; \*\*\*\**P* < .0001. Cer, ceramide; DHCer, dihydroceramide; PhytoCer, phytoceramide; α-OH PhytoCer, α-hydroxylated phytoceramide; GlcCer, glucosylceramide; α-OH GlcCer, α-hydroxylated glucosylceramide; plank, planktonic cells; Pilo, cells treated with PHCl.

Since accumulated ROS causes oxidative damage to essential biomolecules, triggering apoptosis (17), we then evaluated whether the decreased biofilm vitality and biomass production could involve apoptosis. We performed a double staining using APC-conjugated Annexin V (AnV) antibody, which binds to the cells that manifest phosphatidylserine on the surface of the membrane, as an apoptotic process biomarker, and propidium iodide (PI) as a necrosis marker, followed by flow cytometry analysis. After 24 hours of incubation with PHCl, we observed a significant increase.

## DISCUSSION

PHCl hydrochloride, a nonspecific muscarinic agonist, has previously been shown to inhibit filamentation and biofilm formation in both *in vivo* and *in vitro C. albicans* models (10). Although it has been suggested that *C. albicans* possesses an uncharacterized cholinergic receptor involved in the regulation of filamentation, the mechanisms responsible for PHCl anti-biofilm activity remain unclear. This study investigated several pathways that could lead to *C. albicans* filamentation, biofilm inhibition, and changes in cell growth.

Given that both lipids and sphingolipids have been implicated in yeast polarized growth and biofilm formation, we sought to explore whether PHCl could affect their cellular content (18). The comparative lipidomic data suggest an important remodeling of lipids during biofilm formation that is completely abrogated by PHCl treatment. As previously described, polar lipid content increases when *C. albicans* grows in filamentous forms and organizes into aggregates or three-dimensional communities (14).

Biofilm formation is also characterized by a higher cell surface hydrophobicity (CSH), which can also predict the strain propensity to switch to a sessile lifestyle (19). CSH has been shown to be related to cell wall composition and to modulate antifungal resistance. Suchodolski et al. demonstrated that hydrophobic *C. albicans* strains are characterized by enrichment in the ergosterol content and a more pronounced resistance to fluconazole also in the planktonic form (20). Consistent with previous findings from our group on the

**UPREGULATED PROTEINS (16)**

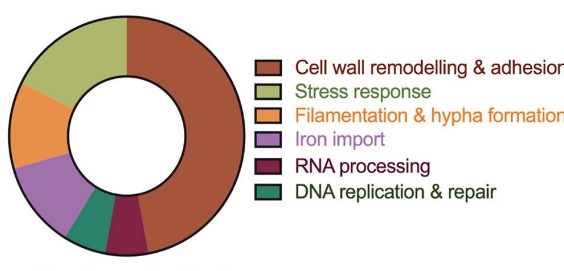

*Sessile vs planktonic*

**DOWNREGULATED PROTEINS (21)**

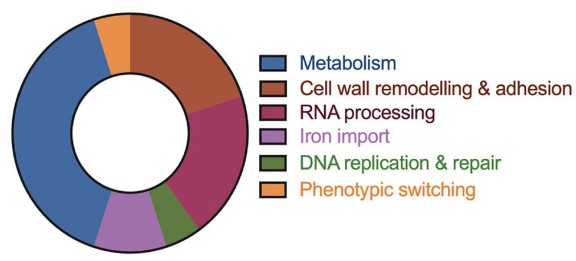

*Sessile vs planktonic*

**UPREGULATED PROTEINS (12)**

*Pilocarpine-treated biofilm vs biofilm*

**DOWNREGULATED PROTEINS (30)**

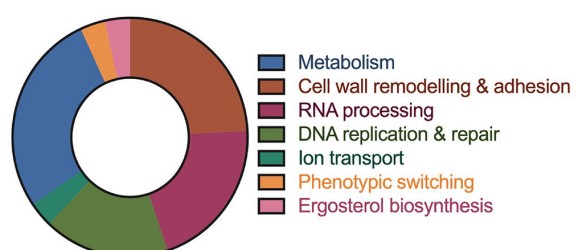

*Pilocarpine-treated biofilm vs biofilm*

**FIG 4** Proteins significantly upregulated and downregulated during biofilm formation (upper donut charts) and by PHCl exposure during biofilm growth (lower donut charts). Proteins were characterized by LC-MS analysis and functional characterization by Proteome Discoverer 2.5 equipped with the *C. albicans* database.

reduction of CSH after 24 hours of PHCl treatment (19), we observed that sessile cells exhibit a higher ergosterol content, which is significantly reduced by PHCl. Ergosterol has been shown to be critical during the early stages of biofilm formation, while its levels decrease in mature communities (21). Our study, conducted on 24-hour-old biofilms, supports the key role of ergosterol, as its depletion by PHCl led to biofilm inhibition. This observation aligns with the findings of Derkacz et al., who reported a link between ergosterol content and filamentous growth. Indeed, changes in ergosterol levels result in substantial alterations in *C. albicans'* ability to form biofilms and undergo filamentation (22).

Ergosterol homeostasis is crucial for proper cell membrane function and fungal virulence and is tightly regulated by several key factors encoded by both essential and nonessential genes (23, 24). Among the nonessential genes is *ERG2*, which encodes a C-8 sterol isomerase responsible for converting fecosterol to episterol (25). This gene is strongly downregulated by PHCl. Consistently, we observed a concurrent accumulation of lanosterol, a precursor of ergosterol. Kodedová et al. reported that the *ERG2* knockout strains are viable but show disruption of ergosterol biosynthesis and accumulation of aberrant sterols, leading to susceptibility to stress agents (26). Using the *C. albicans* gene replacement and conditional expression (GRACE) collection, O'Meara et al. showed that *ERG2* mutants have a hyphal growth defect, as we observed in PHCl-treated biofilms (27). Previous research has demonstrated that targeting the Erg2 enzyme could be an effective strategy to reduce *C. albicans* invasiveness and treat disseminated infections (28).

Ergosterol is a well-established target for antifungal drugs, with its biosynthesis being the primary target of azoles, one of the most widely used antifungal classes (29). By modulating Erg2 activity, one of PHCl's mechanisms of action appears to mirror azole antifungal effects.

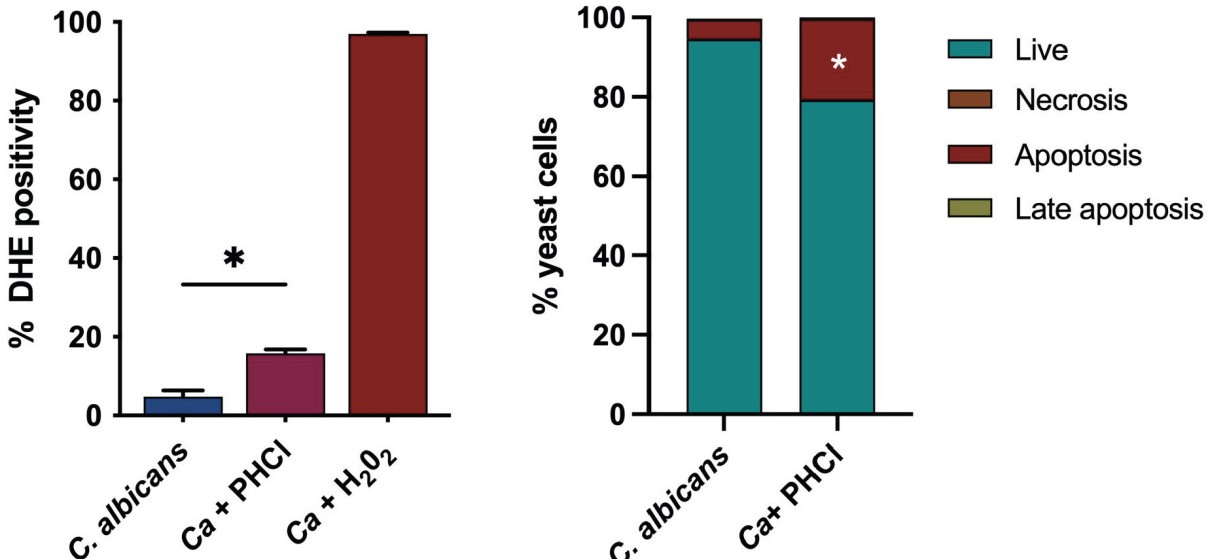

**FIG 5** PHCl effects on reactive oxygen species production and apoptosis in *C. albicans*. PHCl 25 mM treatment increases the percentage of yeast cells that produce ROS (panel A). Dihydroethidium (DHE) fluorescence was measured by flow cytometry. Detection of apoptosis and necrosis (panel B) using Annexin V-APC and PI dual staining. The histogram shows the percentage of live (Annexin V and PI negative), early apoptotic (Annexin V-positive, PI negative), late apoptotic (Annexin V-positive, PI positive), and necrotic (Annexin V-negative, PI positive) *C. albicans* cells (control and PHCl treated). Statistical analysis by Mann-Whitney test; *$P<0.05$.

According to our experiments, the inhibitory effect of PHCl on *C. albicans* growth extends beyond alterations in lipid content.

Within the plasma membrane, there is a complex association between sterols, sphingolipids, and proteins that bestows cell membranes' specific functions (30). Discrete membrane microdomains within lipid bilayers, known as lipid rafts, are enriched in sphingolipids and sterols (31). These specialized regions enable a nonrandom membrane architecture designed to facilitate and organize specific functional processes (30). Hence, variations in the sphingolipids (SLPs), lipids, or protein content, as well as their interactions (lipid-lipid and lipid-protein) in *Candida* membrane, could lead to functional alterations that, in turn, can modulate fungal virulence.

Besides being critical components of lipid rafts, SLPs play an essential role as signaling molecules in various yeast processes, including cell wall remodeling and drug resistance/tolerance (32, 33).

SLP biosynthesis is highly conserved among eukaryotes, and inhibition of the common steps between fungi and mammals, for example, by myriocin, has been shown to block the growth of fungi such as *Candida* spp. and *Aspergillus fumigatus* (34, 35).

As *Candida* sessile cells are physiologically distinct from their planktonic counterpart, we investigated the differences in SLPs according to the growth mode.

PHCl treatment significantly affected glucosylceramide and phytoceramide levels, with both planktonic and sessile yeast cells showing drug-induced enrichment. Additionally, treated biofilms exhibited increased alpha-hydroxylated forms.

The hydroxylation of ceramide and sphingolipids has been reported to alter their cellular location and membrane physical properties (36), but the physiological role of the different hydroxylation states is still largely unknown.

GlcCer are pivotal for virulence, but their accumulation in the cytoplasm can reduce fungal filamentation and the ability to colonize the host (37). Among the reported functions of GlcCer is its correlation with lipid rafts and secretion, as well as its biosynthetic steps as potential targets for new antifungal drugs (38).

The significant accumulation of GlcCer induced by PHCl treatment suggests potential cellular stress, prompting the activation of protective mechanisms. Notably, sphingolipid

accumulation has been reported to cause cell death in both filamentous fungi and *Candida*, as described by Mingione et al. and Thevissen et al., respectively (39, 40). PHCl treatment induced the accumulation of PhytoCer, particularly in sessile cells, which has been linked to fungal cell cycle arrest and apoptosis (41). Disruption of IPC synthesis from PhytoCer impairs fungal cells' ability to infect host tissues, making them significantly more susceptible to immune attack (42). Consequently, IPC synthase has emerged as a promising target for antifungal drug development (43).

Apoptosis, in turn, is considered a novel antifungal mechanism, as fungicidal compounds promote ROS-dependent cell death through a shared signaling and metabolic cascade (44). Furthermore, ROS act as essential intracellular messengers during yeast apoptosis (45).

PHCl has been reported to induce apoptosis and ROS production in human hepatoma-derived Hep G2 cells (16).

However, its effects on other eukaryotic cells, such as *C. albicans*, remain unexplored. Our data indicate that PHCl can induce apoptosis, likely through ROS production, which was significantly elevated in our planktonic model.

The proteomic data did not show overexpression of proteins involved in the apoptotic pathway, such as the oxidative stress-related genes *Rhr2*, *Adh7*, *and Ebp1*, nor in metacaspase activation, i.e., *Mca1* (46).

Nevertheless, cell wall and membrane alterations are known to trigger ROS production that causes DNA damage leading to apoptosis (47).

Regardless of apoptosis-related proteins, hyphal-associated proteins, i.e., Als3, Hyr1, and Cht2, were increased in untreated biofilms compared to planktonic cells, whereas only Rfx2, among the key regulators of biofilm production, was upregulated in biofilms compared to planktonic cells or in biofilms treated with PHCl. *Rfx2* is the only negative regulator among the nine master transcription factors and is expressed at intermediate time points during hyphal and biofilm formation, particularly at 24 hours (48). This could partially explain our data, as our analysis focused on such a time point.

PHCl treatment caused the biofilm proteome to share several features with its planktonic counterpart. Eng1, Rbe1, Ywp1, and Als4 were exclusively found in PHCl-treated sessile cells. This observation aligns with a previous study on the *C. albicans* secretome across various growth conditions and morphological forms, which showed significant differences in secreted proteins between yeast and hyphal cultures (49).

A limitation of this study is that all experiments were performed using only the reference strain *C. albicans* SC5314, which may not fully capture the diversity of behaviors exhibited by other isolates, particularly clinical ones.

However, *C. albicans* SC5314 is well characterized for its ability to form biofilms and provided valuable preliminary evidence that PHCl treatment interferes with *C. albicans* biofilm formation, filamentation, and viability through multiple mechanisms, including lipid reorganization, ergosterol depletion, and sphingolipid accumulation.

In conclusion, the results of this pilot study warrant further investigation across multiple *C. albicans* strains to generalize the results and assess the potential of PHCl as a therapeutic agent against *C. albicans* infections.

## ACKNOWLEDGMENTS

The metabolomic analysis was carried out at the Unitech OMICs platform, the mass spectrometry facility of the University of Milano (Milan, Italy). The authors acknowledge the support of the University of Milan through the APC initiative.

## AUTHOR AFFILIATIONS

[1]Department of Health Sciences, Università degli Studi di Milano, Milan, Lombardy, Italy
[2]Department of Biomedical, Surgical and Dental Sciences, Università degli Studi di Milano, Milan, Lombardy, Italy

## AUTHOR ORCIDs

Emerenziana Ottaviano http://orcid.org/0000-0003-3839-2698
Michele Dei Cas http://orcid.org/0000-0001-7359-8558
Silvia Ancona http://orcid.org/0000-0003-1999-129X
Francesca Triva http://orcid.org/0009-0005-5785-6795
Sara Casati http://orcid.org/0000-0002-5049-7321
Francesca Sisto http://orcid.org/0000-0003-0166-2164
Elisa Borghi http://orcid.org/0000-0002-1893-0455

## AUTHOR CONTRIBUTIONS

Emerenziana Ottaviano, Data curation, Formal analysis, Investigation, Writing – original draft | Michele Dei Cas, Data curation, Formal analysis, Methodology | Silvia Ancona, Investigation, Methodology | Francesca Triva, Investigation | Sara Casati, Formal analysis, Methodology | Francesca Sisto, Formal analysis, Writing – review and editing | Elisa Borghi, Conceptualization, Funding acquisition, Writing – original draft, Writing – review and editing

## DATA AVAILABILITY

Data presented throughout the study are fully available and without restriction, at https://massive.ucsd.edu/ProteoSAFe/dataset.jspaccession=MSV000097355 and on request to the listed authors.

## ADDITIONAL FILES

The following material is available online.

Open Peer Review

**PEER REVIEW HISTORY (review-history.pdf).** An accounting of the reviewer comments and feedback.

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
