## [Reviewer comments · Microbiology Spectrum]

Microbiology Spectrum

Pilocarpine inhibits *Candida albicans* SC5314 biofilm maturation by altering lipid, sphingolipid, and protein content.

Emerenziana Ottaviano, Michele Dei Cas, Silvia Ancona, Francesca triva, Sara Casati, Francesca Sisto, and Elisa Borghi

Corresponding Author(s): Elisa Borghi, Universita degli Studi di Milano

Review Timeline:

Submission Date:	December 19, 2024
Editorial Decision:	January 27, 2025
Revision Received:	February 5, 2025
Accepted:	February 20, 2025

Editor: Eva Pericolini

Reviewer(s): Disclosure of reviewer identity is with reference to reviewer comments included in decision letter(s). The following individuals involved in review of your submission have agreed to reveal their identity: Walid Salem Abu Rayyan (Reviewer #2)

Transaction Report:

DOI: <https://doi.org/10.1128/spectrum.02987-24>

Re: Spectrum02987-24 (Pilocarpine inhibits *Candida albicans* biofilm maturation by altering lipid, sphingolipid, and protein content.)

Dear Prof. Elisa Borghi:

Thank you for the privilege of reviewing your work. Below you will find my comments, instructions from the Spectrum editorial office, and the reviewer comments.

Revision Guidelines

Sincerely,
Eva Pericolini
Editor
Microbiology Spectrum

Reviewer #1 (Comments for the Author):

The manuscript presents an interesting chemical comparison of biofilm-forming and planctonic *C. albicans*. However, the results and conclusions come from only 3 repetitions of the experiment for one *C. albicans* strain. I think this is not enough to draw any conclusions since it can differ among the strains and it should be tested. The wording should be revised to address this and I would recommend the authors to call it a pilot study.

I have some more quick notes:

YPD, PBS and RPMI origin

Shaking incubator, thermomixer and centrifuge details

The details on centrifugation should be all in „g“, if rpm is used, the rotor must be specified

Point-by-point reply

We sincerely thank Reviewer #1 for appreciating our manuscript and for the valuable comments, all of which we have fully addressed.

The manuscript presents an interesting chemical comparison of biofilm-forming and planctonic *C.albicans*. However, the results and conclusions come from only 3 repetitions of the experiment for one *C. albicans* strain. I think this is not enough to draw any conclusions since it can differ among the strains and it should be tested. The wording should be revised to address this and I would recommend the authors to call it a pilot study.

We agree with the reviewer that using only the *C. albicans* SC5314 strain may not fully capture the diversity of behaviors exhibited by all *C. albicans* isolates. In response to this suggestion, we have specified the strain in the paper's title and in the text, and included a limitations paragraph in the discussion section, emphasizing the pilot nature of our findings.

I have some more quick notes:

- YPD, PBS, and RPMI origins
We have included this information in the Materials & Methods section.
- Details on the shaking incubator, thermomixer, and centrifuge
We have provided the relevant specifications in the Materials & Methods section.
- Centrifugation details should be in "g"; if rpm is used, the rotor must be specified
We have specified the centrifuge model and presented the protocol details in "g" where applicable. However, in lines 93 and 107, we have retained the rpm measurement, as these apparatuses are a shaker incubator and an oscillator mixer, respectively.

Re: Spectrum02987-24R1 (Pilocarpine inhibits *Candida albicans* SC5314 biofilm maturation by altering lipid, sphingolipid, and protein content.)

Dear Prof. Elisa Borghi:

Your manuscript has been accepted, and I am forwarding it to the ASM production staff for publication. Your paper will first be checked to make sure all elements meet the technical requirements. ASM staff will contact you if anything needs to be revised before copyediting and production can begin. Otherwise, you will be notified when your proofs are ready to be viewed.

Sincerely,
Eva Pericolini
Editor
Microbiology Spectrum

Reviewer #1 (Comments for the Author):

The authors have addressed all my concerns.